# Cardiovascular disease risk in liver transplant recipients transplanted due to chronic viral hepatitis

**Paolo Maggi**[1]*, **Federica Calò**[1], **Vincenzo Messina**[2], **Gianfranca Stornaiuolo**[1], **Maria Stanzione**[1], **Luca Rinaldi**[3], **Stefania De Pascalis**[1], **Margherita Macera**[1], **Nicola Coppola**[1]

1 Department of Mental Health and Public Medicine – Infectious Diseases Unit, University of Campania Luigi Vanvitelli, Naples, Italy, 2 Infectious Disease Unit, AORN Caserta, Caserta, Italy, 3 Department of Advanced Medical and Surgical Sciences, University of Campania Luigi Vanvitelli, Naples, Italy

* paolo.maggi@unicampania.it

**Data Availability Statement:** All relevant data are within the manuscript.

**Funding:** The authors received no specific fundings for this work.

## Abstract

### Background

Cardiovascular disease (CVD) is a major cause of morbidity and mortality after liver transplantation, mostly in patients transplanted for nonalcoholic steatohepatitis, obesity and diabetes. Few data exist on cardiovascular diseases among patients transplanted for viral hepatitis.

### Objective

Our aim is to clarify the cardiovascular risk and subclinical vascular damage among liver transplant recipients for chronic viral hepatitis (i.e. hepatits C virus, hepatis B virus and hepatitis D virus infection).

### Methods

Adult patients (age ≥ 18 years) with orthotopic liver transplants (OLT) due to viral hepatitis who signed informed consent, and were admitted for a routine follow-up between June 2019 and September 2020 at the Infectious Disease outpatient clinic of the University of Campania Luigi Vanvitelli, Naples, Italy, were prospectively enrolled. An estimation of cardiovascular risk was assessed using three main risk charts, echocolor-Doppler of epiaortic vessels was performed to assess subclinical Intima-Media changes.

### Results

A total of 161 patients were evaluated; of these 15 were excluded because not affected by viral hepatitis. 146 patients were considered. 83 patients (56.8%) were considered at high cardiovascular risk according to Framingham, 54 patients (36.9%) to American Heart Association Arteriosclerotic Cardiovascular Disease (ASCVD) score and 19 (13.0%) to Heart Score. Only 8 patients (5.4%) showed a normal carotid ultrasound, while 52 patients

**Competing interests:** NO authors have competing interests.

(35.6%) had a carotid artery Intima-Media Thickness (IMT) and 86 (58.9%) an atherosclerotic plaque.

## Conclusions

Liver transplant recipients for virus-related associated liver disease are, in light of the high percentage of carotid lesions, at high risk of CVD. Risk charts compared to subclinical carotid lesions which represent damage already established and a real localization of the disease, seem to underestimate the cardiovascular risk. A chronic inflammatory status, could play a key role. It's important to raise the awareness of cardiovascular risk in liver transplant patients to prevent cardiovascular diseases and improve the timing of early diagnosis of premature vascular lesions.

## Introduction

Liver transplantation (LT) is a lifesaving procedure for patients with acute and chronic end stage liver disease, and hepatocellular carcinoma [1–3].

Hepatotrophic viruses capable of causing chronic hepatitis (i.e. hepatitis C virus, hepatitis B virus and hepatitis D virus) are among the most common causes of cirrhosis and a frequent indication for liver transplantation, in particular among individuals with co-infections who are at increased risk of end stage liver disease.

Nowadays, LT is a routine procedure with excellent outcomes in terms of quality and length of life. However, complications are frequent both in the early and long-term period, and significantly impact in terms of morbidity and mortality [4–8]. In particular, cardiovascular complications and metabolic alterations among liver transplant recipients, represent, the main early and late complications during follow-up [9].

The cumulative cardiovascular risk events are estimated between 13.6% and 42%, depending on the studies [10, 11]. The cause of this reported high risk is multifactorial, including, on one side, chronic immunosuppressive treatment [12] and, on the other side, end-stage liver disease that leads to metabolic disturbances of lipids glucose, and blood pressure levels. As a matter of fact, it is well known that approximately half of the cirrhotic patients suffer for associated cardiomyopathy characterized by high heart rate, increased cardiac output, and decreased systemic vascular resistance [13, 14]. Moreover, cardiac dysfunction also depends from acute hemodynamic changes caused by reperfusion of the graft and the release of proinflammatory cytokines, which substantially affect early cardiovascular risk [15].

Chronic immunosuppressive therapy also represents an important and well recognized risk factor for long-term morbidity due to its influence on dyslipidemia, diabetes mellitus, ischemic heart disease and hypertension [16].

Among calcineurin inhibitors (CNIs), cyclosporine has a greater impact on metabolic and cardiovascular comorbidities than tacrolimus [9, 17–19], while the mammalian target of rapamycin (mTOR) inhibitors lower the risk of metabolic syndrome compared to CNIs, except for their impact on fat metabolism [20].

Moreover, etiology has a major impact on cardiovascular risk, being higher in patients transplanted for nonalcoholic steatohepatitis (NASH), alcoholic cirrhosis, obesity and diabetes- related cause of end-stage liver disease [21].

Although a potential association between hepatitis C virus (HCV) infection and risk of sub-clinical and clinical cardiovascular disease (CVD) has been described, with a 2.5–3.5% absolute risk increase of 10-year CVD in HCV-infected subjects compared to the HCV-negative population [21, 22], few data exist on CVD among stable patients transplanted for viral hepatitis due to different etiology.

In this scenario, the objective of the present study was to evaluate the risk of cardiovascular disease and the presence of subclinical vascular damage among liver transplant recipients infected with HCV and/or hepatitis B virus (HBV) and hepatitis D virus (HDV)-coinfection associated liver disease, in order to raise the awareness of cardiovascular risk in LT patients, to prevent CVD and improve the timing of early diagnosis of premature vascular lesions.

## Materials and methods

### Study design and patient selection

This is a cross-sectional, observational cohort study including patients with orthotopic liver transplantation (OLT) consecutively admitted for routine follow-up visit to the Infectious Disease outpatient facilities of the University of Campania Luigi Vanvitelli, Naples, Italy from June 2019 to September 2020.

All adult (age ≥ 18 years) patients with liver transplantation due to viral hepatitis who signed informed consent were included; therefore, liver transplant recipients due to other etiologies were excluded.

The study was approved by the Ethics Committee of the Hospital of Campania Luigi Vanvitelli. All procedures in the study protocol were carried out in accordance with the Helsinki Declaration of 1975. Written consent was obtained.

### Variables and definitions

Data collected from all patients included: demographics; etiology; presence of underlying chronic disease; immunosuppression; other chronic therapies. Data regarding independent risk factors for CVD (family history, smoke, alcohol consumption) were also collected.

Each patient underwent a complete physical examination with measurement of blood pressure, weight, height and abdominal circumference as well as routine biochemical tests.

Familial history of cardiovascular disease was defined as a first degree relative-parent, sibling or child that has history of myocardial infarction or ischemic stroke at ages younger than 55 years in men and younger than 65 years in women [23].

Definition of previous cardiovascular diseases included: ischemic heart disease (heart attack and angina pectoris), and cerebrovascular diseases (ischemic and hemorrhagic stroke).

An abdominal obesity was defined as waist circumference greater than 102 cm in male and 88 in female; hypertriglyceridemia as triglycerides values greater than150 mg/dl or when the patient was under specific therapy. Hypertension was defined as blood pressure greater than 130/85 mmHg.

Metabolic syndrome (MS) was defined by the presence (or treatment) of 3 or more of the following [24]:

1. Obesity (waist circumference: men >102 cm, women >88 cm)

2. Fasting plasma glucose ≥ 100 mg/dL (5.6 mmol/L)

3. Blood pressure ≥ 130/85 mm Hg

4. Triglycerides ≥ 150 mg/mL (1.7 mmol/L)

5. Low HDL cholesterol (men<40 mg/dl, women <50 mg/dl)

The three cardiovascular risk charts evaluated were: Framingham risk score [25], American Heart Association Arteriosclerotic Cardiovascular Disease (ASCVD) [26] and the Heart score [27]. We used three different risk scores to distinguish two different outcomes. In fact, Heart Score evaluates the risk of fatal major adverse cardiovascular events, while ASCVD and the Framingham risk score include non-fatal events.

As regards the echocolor-Doppler of epiaortic vessels, ultrasonography was performed by a physician specifically trained on carotid vessels images with 20 years' experience in echocolor-Doppler ultrasound (MP), using a power echocolor-Doppler instrument with 7.5 MHz probes. An Intima-Media Thickness [IMT] ≥1 mm, of common and internal carotid for both left and right sides, was considered pathological. A carotid was classified as being affected by athero-sclerotic plaque if the localized thickening was ≥1.3 mm.

A minimum of three measurements have been carried out: on the common carotid artery: 1 cm before the carotid bifurcation and at carotid bifurcation; on the internal carotid: 1 cm after the carotid bifurcation and 2 cm after the carotid bifurcation. All images were photo-graphed and properly archived.

## Statistical analysis

Continuous variables were summarized as means and standard deviations, whereas categorical variables were indicated as absolute and relative frequencies. CI 95% Clopper–Pearson and binomial test were used when appropriate.

## Results

The demographic and clinical characteristics of enrolled patients are summarized in Table 1. Of the 161 patients with a history of OLT evaluated in the study period, 15 were excluded because not affected by viral hepatitis; thus, an overall of 146 patients were enrolled in the present study (median age 64 years; males 73.9%), of whom 59 (40.4%) were transplanted for HCV, 27 (18.4%) for HBV, 46 (31.5%) for dual infection HBV-HDV, 10 (6.8%) for dual infection HBV-HCV and 4 (2.7%) for triple infection HBV-HDV-HCV end-stage liver disease. For 87 patients (59.5%) the time since transplantation was greater than 10 years, while only 23 (15.6%) patients had been transplanted for less than 5 years. The most frequent immunosuppressive regimen used included tacrolimus as monotherapy (31.6%), followed by the combination of tacrolimus and mycophenolate (18.4%) and cyclosporine alone (14.5%). 36 (24.6%) were active smokers and 19 (13.0%) alcohol users. 58 patients (39.7%) had a familial history for CVD, 75 (51.3%) for diabetes mellitus and 34 (23.3%) for dyslipidemia.

51 (34.9%) patients had diabetes arose in the post-transplant, 44 (30.1%) suffer from dyslipidemia and 11 (7.5%) patients had already experienced an acute cardiovascular event in the form of stroke/transient ischemic attack (TIA).

More than half of the patients (63.1%) were taking antihypertensive therapy, 19.7% a lipid-lowering drug and 32.9% a hypoglycemic therapy in the form of oral antidiabetic drugs or insulin.

Metabolic status and biochemical values are shown in Table 2. At biochemical tests 71 (48.6%) and 52 (35.6%) patients showed, respectively, pathological glycemia and triglycerides values. More than half of the patients (65.8%) showed poor blood pressure control at the visit with values higher than normal. Furthermore, 50 (34.2%) patients had 3 or more criteria for metabolic syndrome.

**Table 1. Demographics and data regarding transplant status and CVD risk n, (%).**

| | |
|---|---|
| All patients | 146 |
| Age (median, range) | 64 (41–80) |
| Male sex, n (%) | 108 (73.9) |
| Caucasian, n (%) | 146 (100) |
| BMI (media) | 29.1 |
| Aetiology | |
| HBV | 27 (18.4) |
| HCV | 59 (40.4) |
| HBV±HDV | 46 (31.5) |
| HBV-HCV | 10 (6.8) |
| HBV-HDV-HCV | 4 (2.7) |
| Time elapsed since transplantation | |
| ≥20 years | 33 (22.6) |
| ≥10 and ≤20 years | 54 (36.9) |
| ≥5 and ≤10 years | 36 (24.6) |
| ≥1 and ≤5 years | 21 (14.3) |
| <1 year | 2 (1.3) |
| Immunosuppressant | |
| Tacrolimus | 46 (31.5) |
| Everolimus | 10 (6.8) |
| Cyclosporine | 21 (14.3) |
| Mycophenolate | 8 (5.4) |
| Tacrolimus plus Mycophenolate | 27 (18.4) |
| Cyclosporine plus Mycophenolate | 15 (10.2) |
| Tacrolimus plus Everolimus | 19 (13.0) |
| Instructions | |
| Elementary schools | 46 (31.5) |
| Middle schools | 58 (39.7) |
| High school | 35 (23.9) |
| University | 7 (4.7) |
| Years of instructions, media (range) | 9.05 (2–18) |
| Working activity | |
| Sedentary | 123 (84.2) |
| Light-moderate | 23(15.7) |
| Physical movement | |
| None | 54.0 (36.9) |
| Usually walks | 86 (58.9) |
| Sport activity | 6 (4.1) |
| Familiarity | |
| CVD | 58 (39.7) |
| Ictus/TIA | 23 (15.7) |
| DM | 75 (51.3) |
| Dyslipidemia | 35 (23.9) |
| Neoplasia | 83 (56.8) |
| Comorbidities | |
| CVD | 8 (5.4) |
| Ictus/TIA | 12 (8.2) |
| DM before LT | 23 (15.7) |

(*Continued*)

**Table 1.** (Continued)

| | |
|---|---|
| DM after LT | 52 (35.6) |
| Dyslipidemia before LT | 0 (0) |
| Dyslipidemia after LT | 44 (30.1) |
| Neoplasia before LT | 54(36.9) |
| Neoplasia after LT | 25 (17.1) |
| Smoker | 36 (24.6) |
| Alcohol users | 19(13.0) |
| Ormonal or corticosteroid therapies | 6 (4.1) |
| Lipid-lowering therapies | 29 (19.9) |
| Antihypertensive therapies | 92 (63.0) |
| Oral antidiabetics | 15 (10.2) |
| Insulin therapy | 33 (22.6) |

BMI: Body massa index, TIA: Transient ischaemic attack, DM: Diabetes mellitus, LT: Liver transplantation.

**Table 2. Metabolic status.**

| Variables | N, (%) |
|---|---|
| Abdominal Obesity | 25 (17.1) |
| Hypertriglyceridemia | 52 (35.6) |
| Low HDL cholesterol | 67 (45.8) |
| Hypertension | 90 (65.8) |
| Hyperglycemia | 71 (46.8) |
| Metabolic syndrome | 50 (34.2) |

As regards the cardiovascular risk charts (Table 3), an overall of 83 patients (56.8%) were considered at high cardiovascular risk according to Framingham, 54 patients (36.9%) to ASCVD and 19 (13.0%) to Heart Score.

Analyzing the echocolor-Doppler features of the patients enrolled (Table 4), only 8 patients (5.4%) showed a normal carotid ultrasound. 138 (94.5%) patients showed pathological ultrasonographic findings; of these, 86 (58.9%) had atherosclerotic plaque and 52 (35.6%) had abnormal IMT (CI 95% Clopper–Pearson = 29,58%–46,32%; binomial test: p = 0.0048).

51.1% of patients who had a plaque had been transplanted for HCV, 22.2% for co-infection HBV±HDV and 17% for HBV. IMT was more frequently found in patients transplanted for co-infection HBV±HDV (44.4%), followed by HCV (25.9%) and HBV (18.5%) monoinfection.

**Table 3. Cardiovascular risk charts.**

| Cardiovascular Risk | Framingham | ASCVD | Heart-SCORE |
|---|---|---|---|
| *Low risk* | 35 (23.8) | 15 (10.3) | 8(5.5) |
| *Borderline* | - | 13 (9.0) | - |
| *Intermediate* | 29 (19.7) | 63(43.5) | - |
| *Moderate Risk* | - | - | 117 (80.1) |
| *High risk* | 83 (56.5) | 54(37.2) | 19 (13.0) |
| *Very high risk* | - | - | 2 (1.4) |

**Table 4. Echocolor-Doppler features.**

| Normal n,% | IMT (≥1) n,% | Plaque (≥1,3) n,% |
|---|---|---|
| 8 (5.4) | 52 (35.6) | 86 (58.9) |

Regarding the correlation between immunosuppressive regimen and echocolor-Doppler features, 31% of patients showing plaques were treated with tacrolimus as monotherapy, followed by the association of tacrolimus plus mycophenolate (15.6%), tacrolimus plus everolimus (15.6%), cyclosporine as monotherapy (15.6%) or combined with mycophenolate (15.6%) and everolimus as monotherapy (6.6%).

## Discussion

Cardiovascular disease has emerged as the leading cause of non-graft-related deaths [28, 29]. In fact, LT represents a major stress which can affect and worsen preexisting cardiovascular dysfunctions [9, 12], mostly in early post-transplant follow-up. Studies concerning long-term cardiovascular complications are fewer than those focused on early morbidity.

The risk of CVD in liver transplant recipients is due to several mechanisms: aging, systemic hemodynamic changes due to advanced liver disease, chronic inflammation and immunosuppressive treatment.

A large number of studies suggest an increased risk of cardiovascular disease events in patients transplanted for nonalcoholic steatohepatitis (NASH), alcoholic cirrhosis, obesity and diabetes-related causes of end-stage liver disease [21]. In fact, a large multicenter cohort study including 32,810 patients with LT showed that major cardiovascular events at 30 and 90 days were independently predicted by age (more than 65 years), pre-transplant creatinine, pretransplant comorbidities such as atrial fibrillation and stroke, and non-viral liver disease such as alcoholic cirrhosis, and non-alcoholic steatohepatitis [21].

The association between CVD and HCV infection in subjects without LT was already known. In fact, a meta-analysis of aggregate data from 22 studies showed that, compared with uninfected controls, HCV-infected patients are at increased risks of CVD-related mortality, carotid plaques and cerebrocardiovascular events [30]. However, despite these data, few studies explore cardiovascular risk and subclinical damage among stable patients transplanted for viral hepatitis.

In the present study, we evaluated the cardiovascular risk and the subclinical damage of the carotid vessels among liver transplant recipients for HCV, HBV±HDV associated liver disease. Until now, cardiovascular assessment among these patients has been poorly investigated, also because they are not perceived at increased risk, differently from patients transplanted for non-alcoholic steatohepatitis (NASH), alcoholic cirrhosis, obesity and diabetes. Unexpectedly, our data suggested that this category of patients are at high cardiovascular risk. In fact, according to Framingham risk score, patients at high cardiovascular risk were more than 50%. Analyzing the echocolor-Doppler images, it is noteworthy that 94.7% of the patients showed pathological ultrasonographic findings (IMT or plaques). This shows that Framingham risk score in these subjects performs better than ASCVD and Heart Score, but underestimates the presence of subclinical vascular damage. We hypothesize that, in this phenomenon, inflammatory mechanisms, not explored by the risk scores at present validated, could play a role. Also immunosuppressive therapy could contribute to the possible interplay between immune suppression and immune activation. In this, patients in treatment with tacrolimus seem to have a similar pattern of cardiovascular risk and vessel damage development with respect to the other

immunosuppressive regimens, even if the number of patients does not allow to infer definitive conclusions.

Possible limits of our study are the number of enrolled subjects, and the absence of a control group. An evaluation of inflammatory biomarkers is warranted to better explore the role of chronic endothelial inflammation in this phenomenon.

In conclusion, this is the first study among long term liver transplant recipients for viral hepatitis, revealing an unexpectedly high presence of patients at increased cardiovascular risk, and an even higher number of patients with subclinical epi-aortic vessel lesions. We suggest that echocolor-Doppler investigation of carotid vessels could be introduced among periodic routine controls of liver transplant recipients for HCV, HBV± HDV-associated disease.

## Author Contributions

**Conceptualization:** Paolo Maggi, Nicola Coppola.

**Data curation:** Paolo Maggi, Federica Calò, Vincenzo Messina, Gianfranca Stornaiuolo, Nicola Coppola.

**Formal analysis:** Paolo Maggi.

**Investigation:** Paolo Maggi, Federica Calò, Vincenzo Messina, Gianfranca Stornaiuolo, Maria Stanzione, Stefania De Pascalis, Margherita Macera.

**Supervision:** Paolo Maggi, Nicola Coppola.

**Validation:** Paolo Maggi, Luca Rinaldi.

**Writing – original draft:** Paolo Maggi, Federica Calò, Gianfranca Stornaiuolo.

**Writing – review & editing:** Paolo Maggi, Vincenzo Messina, Luca Rinaldi, Nicola Coppola.

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
