## [Decision Letter · Decision Letter 0]

16 Jul 2021

PONE-D-21-16062

Cardiovascular Disease (CVD) risk in liver transplant recipients for HCV, HBV± HDV-associated liver disease

PLOS ONE

Dear Dr. Maggi,

Thank you for submitting your manuscript to PLOS ONE. After careful consideration, we feel that it has merit but does not fully meet PLOS ONE’s publication criteria as it currently stands. Therefore, we invite you to submit a revised version of the manuscript that addresses the points raised during the review process.

The Editor reviewed the manuscript. This work requires more relevant information to make suggested conclusions.

The Editor's comments:

The title needs to be modified. What is “HCV, HBV+/- HDV-associated liver disease”? This is not obvious or even clear for a professional and especially for a lay reader. The title needs to be better expressing the entire idea. 

In the Abstract, the authors use multiple abbreviations (ASCVD, IMT, CVD) without an explanation; all abbreviations need to be spelled out when used the first time. 

The conclusion sentences in the Abstract need to be re-written as “Liver transplant recipients for HCV, HBV±HDV-associated liver disease are at high risk of CVD. Comparing the high percentage of subclinical carotid lesions with data of the risk charts, the latter seem to underestimate the real extent of the endothelial damage. A chronic inflammatory status could play a key role. It’s important to raise the awareness of cardiovascular risk in liver transplant patients to prevent cardiovascular diseases and improve the timing of early diagnosis of premature vascular lesions” is not clear. The first sentence does not clearly state the main findings. The second sentence also needs to be better expressed. 

It is confusing the use of CV as cumulative cardiovascular risk as well as CVD as clinical cardiovascular disease. What is the difference? It is not convincing why the authors use these two similar terms. Please explain better each term or use one universal term. The risk of “a 10-year CVD risk 2.5-3.5% higher in HCV-infected subjects compared to the HCV-negative population” is not very impressive. If the risk is for example 20% and an increase is to 22.5% such increase may be biologically not significant. The argument is not convincing. Please explain better this argument. 

The last paragraph in the Introduction section needs to be re-written. For example it should be: “The objective of this study was to evaluate the risk of cardiovascular disease or at least subclinical vascular damage among liver transplant recipients infected with HCV and/or HBV.” 

The authors fail to explain the principal idea for their work. After reading the Abstract and the Introduction, the reader still does not understand what is “HBV ± HDV-associated liver disease”. The abbreviations need to be explained and the combination of HBV +/- HDV should be clearly spelled out and explained.   

The best description of this study would be: “This an observational study of patients with orthotopic liver transplants (OLT) admitted for a routine follow-up between June 2019 and September 2020 at the Infectious Disease outpatient clinic of the University of Campania Luigi Vanvitelli, Naples, Italy. All adult (age ≥ 18 years) patients with liver transplants due to viral hepatitis who signed informative consent were included.” Some form of this description should be in the Abstract.

The authors need to provide the list of all abbreviations with spell out text. 

All abbreviations used in the Results section needs to be explained “were transplanted for HCV, 27 (18.4%) for HBV, 46 (31.5%) for dual infection HBV-HDV, 10 (6.8%) for dual infection HBV-HCV and 4 (2.7%) for triple infection HBV-HDV-HCV. For 87 patients (59.5%)”.

The Introduction section must explain very clearly about different types of hepatitis. There is no any information about what is hepatitis. Only after looking at a website, one can learn “Hepatitis is most commonly caused by the viruses hepatitis A, B, C, D and E. The authors need to write a section with an explanation about hepatitis with explained abbreviations. What are the differences among patients with hepatitis B, C and D versus combination of two different types?

The title of this study is about cardiovascular disease risk in liver recipients with hepatitis. There is no convincing argument that these patients are indeed at higher risk than any other group of patients. Please provide more data and the comparison with non-infected patients for their risk for cardiovascular diseases. The statement at the end of the Discussion section is not satisfactory “Possible limits of our study are the number of enrolled subjects, and the absence of a control group.” In addition, the authors state “However, this is the first study among long term liver transplant recipients for viral hepatitis, revealing an unexpectedly high presence of patients at high CVD risk, and an even higher number of patients with subclinical epiaortic vessel lesions.” The argument needs to be supported by the comparison with other groups of patients. 

Reviewer # 1:

The manuscript presents the results of a observational study which assessed cardiovascular risk in long-term liver transplant recipients (most >5 years) transplanted for viral hepatitis using different risk charts and subclinical intima-media damage via carotid ultrasound. Many recipients had high CV risk and the majority had either IMT or a carotid plaque on CD carotid ultrasound.

The study lacks control groups as was observed in the discussion, however the authors did not draw any exaggerated conclusions Only proportions were reported, with no formal statistical testing, which is in line with the study design.

Table 1 stated that 39,7% of patients had previous cardiovascular disease, how was it defined?

As the present cross-sectional study was not designed to record adverse events at all, carotid ultrasound was used as surrogate. Even though it was previously established that IMT correlates with major adverse cardiovascular events, the risk scores were not originally designed to asses endothelial damage. The risk scores used in the study also differ in outcomes; Heart Score only evaluates the risk of fatal major adverse cardiovascular events, while ASCVD and the Framingham risk score include non-fatal events. Limitations regarding differences in outcomes should be noted in the discussion.

Another observation is that the authors stated in their discussion that patients on tacrolimus monotherapy seemed more prone to developing vessel lesions although the proportions reported were similar to those of the total study population (around 31%).

Reviwer # 2:

The paper by Dr. Magii et al. is a description of a brief observational study of a group of liver transplant recipients whose initial liver failure was due to viral etiology. Some questions should be addressed before publication.

1.     What was the medical history of patients before liver transplantation? Please make sure to include summarized and statistically analyzed information on parameters relevant to CVD risk, such as CVD diagnosis, BMI, diet and lifestyle, lipid profiles etc. Please comment on the change in the CVD risk factors before and after transplantation. This is one way to address the lack of a healthy control group in the study.

2.     The statement about correlation between echo-Doppler features and immunosuppression regimen has to be accompanied with statistical confirmation (probably Chi-squared test).

3.     Please make sure percentages in Table 3 add up to 100 (may require checking rounding).

4.     The manuscript will benefit from proofreading to fix language errors.

We look forward to receiving your revised manuscript.

Kind regards,

Stanislaw Stepkowski

Academic Editor

PLOS ONE

Journal Requirements:

2. In the ethics statement in the manuscript Methods and online submission form, please confirm whether consent was informed, and they type of consent obtained (e.g. written/oral.) If oral consent was obtained, please also include the following information:

- Why written consent could not be obtained

- Whether the Institutional Review Board (IRB) approved use of oral consent

- How oral consent was documented

For more information, please see our guidelines for human subjects research: https://journals.plos.org/plosone/s/submission-guidelines#loc-human-subjects-research

Additional Editor Comments:

The Editor reviewed the manuscript. This work requires more relevant information to make suggested conclusions.

The Editors comments:

The title needs to be modified. What is “VCV, HBV+/- HDV-associated liver disease”? This is not obvious or even clear for a professional and especially for a lay reader. The title needs to be better expressing the entire idea.

In the Abstract, the authors use multiple abbreviations (ASCVD, IMT, CVD) without an explanation; all abbreviations need to be spelled out when used the first time.

The conclusion sentences in the Abstract need to be re-written as “Liver transplant recipients for HCV, HBV±HDV-associated liver disease are at high risk of CVD. Comparing the high percentage of subclinical carotid lesions with data of the risk charts, the latter seem to underestimate the real extent of the endothelial damage. A chronic inflammatory status could play a key role. It’s important to raise the awareness of cardiovascular risk in liver transplant patients to prevent cardiovascular diseases and improve the timing of early diagnosis of premature vascular lesions” is not clear. The first sentence does not clearly state the main findings. The second sentence also needs to be better expressed.

It is confusing the use of CV as cumulative cardiovascular risk as well as CVD as clinical cardiovascular disease. What is the difference? It is not convincing why the authors use these two similar terms. Please explain better each term or use one universal term. The risk of “a 10-year CVD risk 2.5-3.5% higher in HCV-infected subjects compared to the HCV-negative population” is not very impressive. If the risk is for example 20% and an increase is to 22.5% such increase may be biologically not significant. The argument is not convincing. Please explain better this argument.

The last paragraph in the Introduction section needs to be re-written. For example it should be: “The objective of this study was to evaluate the risk of cardiovascular disease or at least subclinical vascular damage among liver transplant recipients infected with HCV and/or HBV.”

The authors fail to explain the principal idea for their work. After reading the Abstract and the Introduction, the reader still does not understand what is “HBV ± HDV-associated liver disease”. The abbreviations need to be explained and the combination of HBV +/- HDV should be clearly spelled out and explained.

The best description of this study would be: “This an observational study of patients with orthotopic liver transplants (OLT) admitted for a routine follow-up between June 2019 and September 2020 at the Infectious Disease outpatient clinic of the University of Campania Luigi Vanvitelli, Naples, Italy. All adult (age ≥ 18 years) patients with liver transplants due to viral hepatitis who signed informative consent were included.” Some form of this description should be in the Abstract.

The authors need to provide the list of all abbreviations with spell out text.

All abbreviations used in the Results section needs to be explained “were transplanted for HCV, 27 (18.4%) for HBV, 46 (31.5%) for dual infection HBV-HDV, 10 (6.8%) for dual infection HBV-HCV and 4 (2.7%) for triple infection HBV-HDV-HCV. For 87 patients (59.5%)”.

The Introduction section must explain very clearly about different types of hepatitis. There is no any information about what is hepatitis. Only after looking at a website, one can learn “Hepatitis is most commonly caused by the viruses hepatitis A, B, C, D and E. The authors need to write a section with an explanation about hepatitis with explained abbreviations. What are the differences among patients with hepatitis B, C and D versus combination of two different types?

The title of this study is about cardiovascular disease risk in liver recipients with hepatitis. There is no convincing argument that these patients are indeed at higher risk than any other group of patients. Please provide more data and the comparison with non-infected patients for their risk for cardiovascular diseases. The statement at the end of the Discussion section is not satisfactory “Possible limits of our study are the number of enrolled subjects, and the absence of a control group.” In addition, the authors state “However, this is the first study among long term liver transplant recipients for viral hepatitis, revealing an unexpectedly high presence of patients at high CVD risk, and an even higher number of patients with subclinical epiaortic vessel lesions.” The argument needs to be supported by the comparison with other groups of patients.

Reviewer # 1:

The manuscript presents the results of a observational study which assessed cardiovascular risk in long-term liver transplant recipients (most >5 years) transplanted for viral hepatitis using different risk charts and subclinical intima-media damage via carotid ultrasound. Many recipients had high CV risk and the majority had either IMT or a carotid plaque on CD carotid ultrasound.

The study lacks control groups as was observed in the discussion, however the authors did not draw any exaggerated conclusions Only proportions were reported, with no formal statistical testing, which is in line with the study design.

Table 1 stated that 39,7% of patients had previous cardiovascular disease, how was it defined?

As the present cross-sectional study was not designed to record adverse events at all, carotid ultrasound was used as surrogate. Even though it was previously established that IMT correlates with major adverse cardiovascular events, the risk scores were not originally designed to asses endothelial damage. The risk scores used in the study also differ in outcomes; Heart Score only evaluates the risk of fatal major adverse cardiovascular events, while ASCVD and the Framingham risk score include non-fatal events. Limitations regarding differences in outcomes should be noted in the discussion.

Another observation is that the authors stated in their discussion that patients on tacrolimus monotherapy seemed more prone to developing vessel lesions although the proportions reported were similar to those of the total study population (around 31%).

Reviwer # 2:

The paper by Dr. Magii et al. is a description of a brief observational study of a group of liver transplant recipients whose initial liver failure was due to viral etiology. Some questions should be addressed before publication.

1. What was the medical history of patients before liver transplantation? Please make sure to include summarized and statistically analyzed information on parameters relevant to CVD risk, such as CVD diagnosis, BMI, diet and lifestyle, lipid profiles etc. Please comment on the change in the CVD risk factors before and after transplantation. This is one way to address the lack of a healthy control group in the study.

2. The statement about correlation between echo-Doppler features and immunosuppression regimen has to be accompanied with statistical confirmation (probably Chi-squared test).

3. Please make sure percentages in Table 3 add up to 100 (may require checking rounding).

4. The manuscript will benefit from proofreading to fix language errors.

Reviewers' comments:

Reviewer's Responses to Questions

**Comments to the Author**

1. Is the manuscript technically sound, and do the data support the conclusions?

Reviewer #1: Yes

Reviewer #2: No

2. Has the statistical analysis been performed appropriately and rigorously? 

Reviewer #1: Yes

Reviewer #2: No

3. Have the authors made all data underlying the findings in their manuscript fully available?

Reviewer #1: Yes

Reviewer #2: Yes

4. Is the manuscript presented in an intelligible fashion and written in standard English?

Reviewer #1: Yes

Reviewer #2: No

5. Review Comments to the Author

Reviewer #1: The manuscript presents the results of a observational study which assessed cardiovascular risk in long-term liver transplant recipients (most >5 years) transplanted for viral hepatitis using different risk charts and subclinical intima-media damage via carotid ultrasound. Many recipients had high CV risk and the majority had either IMT or a carotid plaque on CD carotid ultrasound.

The study lacks control groups as was observed in the discussion, however the authors did not draw any exaggerated conclusions Only proportions were reported, with no formal statistical testing, which is in line with the study design.

Table 1 stated that 39,7% of patients had previous cardiovascular disease, how was it defined?

As the present cross-sectional study was not designed to record adverse events at all, carotid ultrasound was used as surrogate. Even though it was previously established that IMT correlates with major adverse cardiovascular events, the risk scores were not originally designed to asses endothelial damage. The risk scores used in the study also differ in outcomes; Heart Score only evaluates the risk of fatal major adverse cardiovascular events, while ASCVD and the Framingham risk score include non-fatal events. Limitations regarding differences in outcomes should be noted in the discussion.

Another observation is that the authors stated in their discussion that patients on tacrolimus monotherapy seemed more prone to developing vessel lesions although the proportions reported were similar to those of the total study population (around 31%).

Reviewer #2: The paper by Dr. Magii et al. is a description of a brief observational study of a group of liver transplant recipients whose initial liver failure was due to viral etiology. Some questions should be addressed before publication.

1. What was the medical history of patients before liver transplantation? Please make sure to include summarized and statistically analyzed information on parameters relevant to CVD risk, such as CVD diagnosis, BMI, diet and lifestyle, lipid profiles etc. Please comment on the change in the CVD risk factors before and after transplantation. This is one way to address the lack of a healthy control group in the study.

2. The statement about correlation between echo-Doppler features and immunosuppression regimen has to be accompanied with statistical confirmation (probably Chi-squared test).

3. Please make sure percentages in Table 3 add up to 100 (may require checking rounding).

4. The manuscript will benefit from proofreading to fix language errors.

6. PLOS authors have the option to publish the peer review history of their article (what does this mean?). If published, this will include your full peer review and any attached files.

Reviewer #1: No

Reviewer #2: **Yes: **Dulat Bekbolsynov

---

## [Author Response · Author response to Decision Letter 0]

25 Nov 2021

Dear Editor,

We re-submit our paper “Cardiovascular Disease (CVD) risk in liver transplant recipients for HCV, HBV± HDV-associated liver disease” modified according to the suggestions of the editor and of the reviewers.

We have made significant revisions to the original manuscript and we are sure that the comments of the reviewers have improved significantly the final version of the paper. We are convinced that it would be of interest to PLOS ONE.

Kind regards,

Paolo Maggi

POINT-BY-POINT ANSWER TO THE COMMENTS OF THE EDITOR

Point 1: The title needs to be modified. What is “HCV, HBV+/- HDV-associated liver disease”? This is not obvious or even clear for a professional and especially for a lay reader. The title needs to be better expressing the entire idea. 

Answer to point 1: We thank the Editor for this suggestion. We have accordingly modified the title. 

Point 2: In the Abstract, the authors use multiple abbreviations (ASCVD, IMT, CVD) without an explanation; all abbreviations need to be spelled out when used the first time. 

Answer to point 2: We thank the Editor for this suggestion. We accordingly modified the text in the abstract spelled out all the abbreviations.

Point 3: The conclusion sentences in the Abstract need to be re-written as “Liver transplant recipients for HCV, HBV±HDV-associated liver disease are at high risk of CVD. Comparing the high percentage of subclinical carotid lesions with data of the risk charts, the latter seem to underestimate the real extent of the endothelial damage. A chronic inflammatory status could play a key role. It’s important to raise the awareness of cardiovascular risk in liver transplant patients to prevent cardiovascular diseases and improve the timing of early diagnosis of premature vascular lesions” is not clear. The first sentence does not clearly state the main findings. The second sentence also needs to be better expressed. 

Answer to point 3: We thank the Editor for this comment. We accordingly modified the text, and clarified some unclear points by providing better explanations.

Point 4: It is confusing the use of CV as cumulative cardiovascular risk as well as CVD as clinical cardiovascular disease. What is the difference? It is not convincing why the authors use these two similar terms. Please explain better each term or use one universal term. 

Answer to point 4: We thank the Editor for this comment. We have modified the text accordingly and, when appropriate, use a universal term.

Point 5: The risk of “a 10-year CVD risk 2.5-3.5% higher in HCV-infected subjects compared to the HCV-negative population” is not very impressive. If the risk is for example 20% and an increase is to 22.5% such increase may be biologically not significant. The argument is not convincing. Please explain better this argument. 

Answer to point 5: We thank the Editor for this comment. In the study of Badawi et al. the authors using a multivariable linear regression model found as HCV infection was significantly associated with a 2.5–3.5% absolute risk increase of 10-year CVD compared to HCV negative population. We have accordingly modified the text.

Point 6: The last paragraph in the Introduction section needs to be re-written. For example it should be: “The objective of this study was to evaluate the risk of cardiovascular disease or at least subclinical vascular damage among liver transplant recipients infected with HCV and/or HBV.” 

Answer to point 6: We thank the Editor for this suggestion. We have accordingly modified the last paragraph in the Introduction.

Point 7: The authors fail to explain the principal idea for their work. After reading the Abstract and the Introduction, the reader still does not understand what is “HBV ± HDV-associated liver disease”. The abbreviations need to be explained and the combination of HBV +/- HDV should be clearly spelled out and explained. 

Answer to point 7: We thank the Editor for this comment. We have explained all the abbreviations and we have spelled out the combination of HBV +/- HDV as coinfection.

Point 8: The best description of this study would be: “This an observational study of patients with orthotopic liver transplants (OLT) admitted for a routine follow-up between June 2019 and September 2020 at the Infectious Disease outpatient clinic of the University of Campania Luigi Vanvitelli, Naples, Italy. All adult (age ≥ 18 years) patients with liver transplants due to viral hepatitis who signed informative consent were included.” Some form of this description should be in the Abstract.

Answer to point 8: We thank the Editor for this suggestion. We have accordingly modified the section of methods in the abstract.

Point 9: The authors need to provide the list of all abbreviations with spell out text. All abbreviations used in the Results section needs to be explained “were transplanted for HCV, 27 (18.4%) for HBV, 46 (31.5%) for dual infection HBV-HDV, 10 (6.8%) for dual infection HBV-HCV and 4 (2.7%) for triple infection HBV-HDV-HCV. For 87 patients (59.5%)”.

Answer to point 9: We thank the Editor for this comment. We have already explained all the abbreviations in the Introduction.

Point 10: The Introduction section must explain very clearly about different types of hepatitis. There is no any information about what is hepatitis. Only after looking at a website, one can learn “Hepatitis is most commonly caused by the viruses hepatitis A, B, C, D and E. The authors need to write a section with an explanation about hepatitis with explained abbreviations. What are the differences among patients with hepatitis B, C and D versus combination of two different types?

Answer to point 10: We thank the Editor for this comment. We have added this information to lines 66-69 of the new manuscript.

Point 11: The title of this study is about cardiovascular disease risk in liver recipients with hepatitis. There is no convincing argument that these patients are indeed at higher risk than any other group of patients. Please provide more data and the comparison with non-infected patients for their risk for cardiovascular diseases. The statement at the end of the Discussion section is not satisfactory “Possible limits of our study are the number of enrolled subjects, and the absence of a control group.” In addition, the authors state “However, this is the first study among long term liver transplant recipients for viral hepatitis, revealing an unexpectedly high presence of patients at high CVD risk, and an even higher number of patients with subclinical epiaortic vessel lesions.” The argument needs to be supported by the comparison with other groups of patients. 

Answer to point 11: The Editor is right but unfortunately, we have not explored the impact of cardiovascular disease risk in other groups of patients. This limit was stated in the Discussion section; surely, it can be interesting to investigate it for a further development of the research. However, it should be noted that there is an important literature supporting the fact that patients with viral hepatitis are at high cardiovascular risk and that subclinical carotid atherosclerosis are asymptomatic stepping stones to clinical CVD. Therefore, considering the consolidated data, our goal was to investigate the cardiovascular risk in this particular category of patients, i.e. patients undergoing liver transplantation for viral hepatitis, and we found a impressively high presence of patients with subclinical epiaortic vessel lesions and high cardiovascular risk according to risk charts. Fe feel this is a relevant datum worth to be communicated and furtherly studied.

Reviewer # 1:

Table 1 stated that 39,7% of patients had previous cardiovascular disease, how was it defined?

RE: Thank you for this suggestion We specified this definition in the method section.

As the present cross-sectional study was not designed to record adverse events at all, carotid ultrasound was used as surrogate. Even though it was previously established that IMT correlates with major adverse cardiovascular events, the risk scores were not originally designed to asses endothelial damage. The risk scores used in the study also differ in outcomes; Heart Score only evaluates the risk of fatal major adverse cardiovascular events, while ASCVD and the Framingham risk score include non-fatal events. Limitations regarding differences in outcomes should be noted in the discussion.

RE: Thank you for these observations. In the discussion we have argued them as limit of the study.

Another observation is that the authors stated in their discussion that patients on tacrolimus monotherapy seemed more prone to developing vessel lesions although the proportions reported were similar to those of the total study population (around 31%).

Re: Thank you for this comment. We corrected this discrepancy in the text. 

Reviwer # 2:

1. What was the medical history of patients before liver transplantation? Please make sure to include summarized and statistically analyzed information on parameters relevant to CVD risk, such as CVD diagnosis, BMI, diet and lifestyle, lipid profiles etc. Please comment on the change in the CVD risk factors before and after transplantation. This is one way to address the lack of a healthy control group in the study.

Re: thank you for this observation. We agree that these data can provide a further starting point for discussion and enrichment of the manuscript. However, most patients are long term transplantation recipients. For this reason, variables such as BMI, lipid, diet style and others, have changed over time (20 years and more, in most cases!) and cannot be useful for a direct comparison between pre and post transplantation

2. The statement about correlation between echo-Doppler features and immunosuppression regimen has to be accompanied with statistical confirmation (probably Chi-squared test).

Re: Thank you for this comment. The Chi-squared test was done. The Method section was corrected accordingly. 

3. Please make sure percentages in Table 3 add up to 100 (may require checking rounding).

Re: We apologize for the mistake. We corrected it.

4. The manuscript will benefit from proofreading to fix language errors.

Re: Thank you for the suggestion. The manuscript was revised by a native English speaker.

We thank the Editor and the reviewers for helping us to improve our paper.

We hope that the paper is now worthy of publication in PLOS ONE

Best regards,

Prof Paolo Maggi

---

## [Decision Letter · Decision Letter 1]

28 Feb 2022

Cardiovascular Disease risk in liver transplant recipients for chronic viral hepatitis

PONE-D-21-16062R1

Dear Dr. Maggi,

We’re pleased to inform you that your manuscript has been judged scientifically suitable for publication and will be formally accepted for publication once it meets all outstanding technical requirements.

Kind regards,

Vanessa Carels

Staff Editor

PLOS ONE

Additional Editor Comments (optional):

Please modify the title to ensure that it is meeting PLOS’ guidelines (https://journals.plos.org/plosone/s/submission-guidelines#loc-title). In particular, the title should be "specific, descriptive, concise, and comprehensible to readers outside the field" and in this case we recommend the reviewer's suggestion "Cardiovascular Disease risk in liver transplant recipients transplanted due to chronic viral hepatitis"

Reviewers' comments:

Reviewer's Responses to Questions

**Comments to the Author**

1. If the authors have adequately addressed your comments raised in a previous round of review and you feel that this manuscript is now acceptable for publication, you may indicate that here to bypass the “Comments to the Author” section, enter your conflict of interest statement in the “Confidential to Editor” section, and submit your "Accept" recommendation.

Reviewer #1: All comments have been addressed

Reviewer #2: All comments have been addressed

2. Is the manuscript technically sound, and do the data support the conclusions?

Reviewer #1: Yes

Reviewer #2: Yes

3. Has the statistical analysis been performed appropriately and rigorously? 

Reviewer #1: Yes

Reviewer #2: Yes

4. Have the authors made all data underlying the findings in their manuscript fully available?

Reviewer #1: Yes

Reviewer #2: Yes

5. Is the manuscript presented in an intelligible fashion and written in standard English?

Reviewer #1: Yes

Reviewer #2: Yes

6. Review Comments to the Author

Reviewer #1: (No Response)

Reviewer #2: This is an important study that provides new results about cardiovascular disease risk in patients with liver transplants who had different types of liver infections. I do not have new comments to add on this manuscript, as it appears solid and important to me, and resort to providing my opinion about how the comments by previous reviewers were addressed.

For points addressing the title of the manuscript, I feel like its updated version is still imperfect in terms of grammar. I would recommend changing the title to something like "Cardiovascular Disease risk in liver transplant recipients transplanted due to chronic viral hepatitis".

Regarding point 11 from the Editor, I agree with the authors that measuring cardiovascular disease risk in non-infected patients was beyond the scope of this manuscript. Going back and revising the study to include comparison to non-infected patients would equate to rejecting the manuscript that otherwise has a valid and important message to deliver. Expanding this study to statistically measure the relative risk of cardiovascular complications in infected and non-infected liver transplant patients can be viewed as a follow-up project to this study.

Reviewer 1 - I feel like their comment about the difference between cardiovascular and Framingham and ASCVD risk scores could be addressed better. The Discussion section does contain a few sentences on this matter (rows 231-234), so it can be argued that the manuscript does not need further revision due to this comment.

Reviewer 2 - while the point 1 about previous medical history is valid, I feel like information provided in Table 1 in the context of the authors' note about long follow-up in most of patients is a satisfactory response.

Overall, I believe this manuscript is in the form suitable for publication.

7. PLOS authors have the option to publish the peer review history of their article (what does this mean?). If published, this will include your full peer review and any attached files.

Reviewer #1: No

Reviewer #2: **Yes: **Dulat Bekbolsynov

---

## [Editor Report · Acceptance letter]

2 Mar 2022

PONE-D-21-16062R1 

Cardiovascular Disease risk in liver transplant recipients transplanted due to chronic viral hepatitis 

Dear Dr. Maggi:

I'm pleased to inform you that your manuscript has been deemed suitable for publication in PLOS ONE. Congratulations! Your manuscript is now with our production department. 

Kind regards, 

on behalf of

Dr. Vanessa Carels 

Staff Editor

PLOS ONE